# KRAS-Wild Pancreatic Cancer—More Targets than Treatment Possibilities?

**DOI:** 10.3390/cancers17233769

**Published:** 2025-11-26

**Authors:** Kamila Krupa, Marta Fudalej, Hanna Miski, Emilia Włoszek, Marta Szymczak, Anna Badowska-Kozakiewicz, Aleksandra Czerw, Andrzej Deptała

**Affiliations:** 1Students’ Scientific Organization of Cancer Cell Biology, Department of Oncology Propaedeutics, Medical University of Warsaw, 01-445 Warsaw, Poland; kamila.krupa@student.wum.edu.pl (K.K.); s090695@student.wum.edu.pl (H.M.); s081087@student.wum.edu.pl (E.W.); s088476@student.wum.edu.pl (M.S.); 2Department of Oncological Propaedeutics, Medical University of Warsaw, 01-445 Warsaw, Poland; marta.fudalej@wum.edu.pl (M.F.); anna.badowska-kozakiewicz@wum.edu.pl (A.B.-K.); 3Department of Oncology, National Medical Institute of the Ministry of the Interior and Administration, 02-507 Warsaw, Poland; 4Department of Health Economics and Medical Law, Medical University of Warsaw, 02-091 Warsaw, Poland; aleksandra.czerw@wum.edu.pl; 5Department of Economic and System Analyses, National Institute of Public Health NIH—National Research Institute, 00-791 Warsaw, Poland

**Keywords:** wild-type KRAS, gene fusions, precision oncology, germline mutations, gene amplifications, BRAF mutations

## Abstract

Pancreatic ductal adenocarcinoma is one of the most aggressive cancers and is frequently diagnosed at an advanced stage, when treatment options are limited. In a small proportion of patients, particularly those without KRAS mutations, tumors may be driven by rare genetic alterations such as gene fusions, mutations, amplifications, or inherited mutations in DNA repair genes. These molecular changes could be the target for precision therapies, improving survival and quality of life. We will focus on alterations such as kinase fusion genes, gene amplifications or mutations, microsatellite instability or defective DNA mismatch repair, activated-MAPK presence without a KRAS mutation, germline mutations, and prospects.

## 1. Introduction

Pancreatic cancer (PC) is the twelfth most common malignancy worldwide, with 511,000 new cases in 2022, and the sixth leading cause of cancer-related mortality, with an estimated 476,000 deaths in 2022 [1]. The prognosis of pancreatic ductal adenocarcinoma (PDAC) depends on the stage of diagnosis. Most patients present with advanced disease, limiting curative options and resulting in a 5-year overall survival (OS) rate of only 3–15% [2]. While mortality rates have stabilized in many high-income regions, persistently poor outcomes and limited treatment efficacy have made PDAC a growing public health concern [1,3].

PDAC development involves disruptions in multiple signaling pathways: the epidermal growth factor receptor (EGFR), RAS/RAF/Mitogen-Activated Protein Kinase Kinase (MEK)/Extracellular Signal-Regulated Kinase (ERK), Transforming Growth Factor Beta (TGF-β), Wnt/β-catenin, Phosphoinositide 3-Kinase (PI3K)/Protein Kinase B (AKT)/Mammalian Target of Rapamycin (mTOR), and NOTCH pathways [4]. Although most cases arise from sporadic somatic mutations, about 10% have a hereditary basis. Germline mutations in breast cancer 1 (*BRCA1*), breast cancer 2 (*BRCA2*), ataxia telangiectasia mutated (*ATM*), partner and localizer of BRCA2 (PALB2), cyclin-dependent kinase inhibitor 2A (*CDKN2A*), tumor protein p53 (*TP53*), serine/threonine kinase 11 (*STK11*), and mismatch repair genes (MLH1, MSH2, MSH6, PMS2) significantly increase PDAC risk. Syndromes such as Peutz–Jeghers, familial atypical multiple mole melanoma (FAMMM), and Lynch syndrome confer a particularly high susceptibility [5,6]. Emerging evidence also suggests a role of the gut microbiome and environmental toxin exposure in pancreatic carcinogenesis [7].

Established risk factors include tobacco smoking, high body mass index (BMI), elevated fasting plasma glucose, and Helicobacter pylori infection, while protective associations have been reported with a history of allergies and higher fruit or folate intake [8]. The early recognition of PDAC remains challenging, as many patients present with non-specific symptoms such as pain, new-onset diabetes, or depression, which may precede the diagnosis by months or even years. Additionally, an awareness of paraneoplastic manifestations, including Trousseau’s syndrome, pancreatic disease, panniculitis, and polyarthritis syndrome, as well as dermatomyositis and polymyositis, could improve earlier disease detection and proper prognosis assessment [9].

The current treatment strategies for PDAC include surgical resection, chemotherapy, radiotherapy, targeted therapy, and emerging immunotherapeutic approaches. Surgery remains the only potentially curative option, but it is feasible in less than 20% of patients. In resectable cases, adjuvant chemotherapy can improve survival to 30–50% [10]. Standard systemic regimens for advanced disease include FOLFIRINOX (folinic acid, fluorouracil, irinotecan, and oxaliplatin) and gemcitabine-based combinations [6,11]. Chemotherapy may be administered as neoadjuvant therapy to reduce tumor size, as adjuvant therapy after surgery, or palliatively in non-surgical candidates [12]. However, chemotherapy resistance remains a significant clinical challenge and contributes to poor patient outcomes [13]. Radiotherapy may be combined with chemotherapy in selected locally advanced or borderline resectable cases, although its survival benefit remains uncertain [6]. Targeted therapy applies to a subset of patients with actionable molecular alterations, aiming to block oncogenic signaling, restore tumor suppressor function, or exploit DNA repair deficiencies [14]. The Kirsten Rat Sarcoma Viral Oncogene (KRAS) mutations, present in 90–95% of PDAC cases—most commonly at codon 12—are linked to poorer survival compared with wild-type (WT) KRAS [15]. However, in recent years, the development of mutation-specific inhibitors, such as sotorasib, adagrasib, MRTX1133, ASP308, and pan-RAS inhibitors, such as BI-2865, ADT-007, ADT-1004, and daraxonrasib, has shown promising results in preclinical and clinical trials [16]. SiRNA-based therapeutics targeting KRAS mutations and cancer vaccines have emerged as a viable cancer treatment method [16,17]. Additionally, microbiota-derived approaches are discovering the mechanisms of drug resistance and how it can impact treatment outcomes [17].

In PDAC, standard chemotherapy such as FOLFIRINOX is used for both KRAS-mutated and WT KRAS tumors, but WT KRAS cases more often harbor actionable alterations that enable the use of molecularly matched targeted therapies, whereas KRAS-mutated tumors typically lack such options [18]. The retrospective analysis of 1856 patients with PC showed that those patients with actionable molecular alterations who were treated with matched therapy had a significantly longer median OS than the group of patients receiving unmatched therapies [19]. Together, these findings indicate that PDAC comprises biologically distinct subgroups that may require differentiated therapeutic strategies. However, WT KRAS PDAC remains less well characterized in the literature. Although genomic reports describe recurrent, potentially targetable alterations, the evidence is dispersed and not integrated into a unified clinical perspective.

Therefore, this review concentrates on current targeted therapies in WT KRAS PC, their underlying molecular mechanisms, and potential future developments to improve patient outcomes. We will focus on alterations such as kinase fusion genes, gene amplification or mutations, microsatellite instability or defective DNA mismatch repair, activated-MAPK presence without a KRAS mutation, germline mutations, and prospects.

## 2. Fusion Genes in Pancreatic Cancer

Recent advances in DNA and RNA sequencing have allowed for an in-depth assessment of mutations and rearrangements occurring in solid tumors, and now they may become targets for precision oncology. In non-surgical patients, the evaluation of these mutations depends on the quality of the sample collected. Currently, endoscopic ultrasound (EUS)-guided fine-needle biopsy (FNB) tissue sampling is fundamental. One meta-analysis on EUS-FNB showed that modified wet-suction has the best sensitivity, integrity, and adequacy compared with the dry-suction technique. The slow-pull technique remains a valuable alternative [20]. However, the relatively high false-negative rate of EUS-FNB delays diagnosis and negatively affects survival outcomes. The results from another meta-analysis showed that contrast-enhanced fine-needle biopsy (CH-EUS-FNB) may be superior to standard EUS-FNA, by guiding the ideal target for aspiration, which leads to a higher accuracy, adequacy, and sensitivity in larger lesions (>1.5 cm or >2 cm); however, more trials are needed to validate these methods [21].

Fusion genes occur more often in male patients. Moreover, they appear in 19.8% of acinar neoplasms, which include acinar cell carcinoma and pancreatoblastoma, and in intraductal neoplasms, as in 1.2% of PDAC, mostly with WT status. In pancreatic tumors, the most common fusion genes are B-Raf (*BRAF*), Fibroblast Growth Factor Receptor 2 (*FGFR2*), Rearranged During Transfection (*RET*), and anaplastic lymphoma kinase (*ALK*); however, the ALK, FGFR2, Neuregulin 1 (NRG1), neurotrophic receptor tyrosine kinase (NTRK), Mitogen-Activated Protein Kinase (MET), and RET fusions were seen typically in PDAC [22].

### 2.1. ALK Fusions

The *ALK* gene is normally found and expressed in the central nervous system. However, the fusion of ALK with a 5′ partner near the kinase-encoding region results in dimerization and the constant activation of downstream signaling pathways, including Janus kinase (JAK)-signal transducer and activator of transcription (STAT), PI3K/AKT, and MEK/ERK, by mimicking ligand-induced activation [23,24]. The prevalence of the ALK fusion gene in PC is rare, at 0.16%; however, it increases to 1.3% among patients < 50 years old. Some of these fusions, like Echinoderm Microtubule-Associated Protein-Like 4 (EML4)-ALK and Striatin (STRN)-ALK, may be found in pancreatic tumors. They could be targeted with ALK inhibitors: crizotinib, ceritinib, and alectinib [25]. Moreover, in the previously treated advanced non-small-cell lung cancer (NSCLC) with *ALK* rearrangement, targeted therapy with ALK inhibitors such as crizotinib has shown superiority compared to standard chemotherapy [26]. Currently, after comprehensive genomic profiling, ALK inhibitors are routinely applied in patients with ALK-rearranged NSCLC.

The comprehensive genomic profiling of 3710 PDAC samples identified only five cases harboring gene translocations without concomitant KRAS mutations. Moreover, they were detected in younger patients, <50 years old. Due to similarities to ALK-positive lung adenocarcinomas, the equivalent response to treatment was possible. Three out of four patients with PDAC and ALK translocations showed a radiographic response, stable disease, and/or normalized blood CA19-9 after ALK inhibitor treatment [25]. One case study described a 34-year-old man with confirmed EML4–ALK fusion KRAS-WT PC, who had a remarkable response to crizotinib after a resistance to prior chemotherapy and re-response to alectinib following the development of brain metastases [27]. Moreover, novel fusion PPFIA-binding protein 1 (PPFIBP1)-ALK was detected in a 41-year-old woman with metastatic PDAC. She responded to alectinib as a first-line targeted therapy and after progression, and acquired alectinib-resistant *ALK* mutations G1202R and V1180L; the disease was treated with a third-generation ALK inhibitor—lorlatinib. At the time of report, the patient survived 10 months and was still alive [28].

Given the limited data on the efficacy of ALK inhibitors in pretreated patients with ALK fusion-positive gastrointestinal (GI) cancers, an international data set and molecular case series of GI tumors were analyzed. In the group of thirteen patients included in the study, five of them had PDAC and one was not evaluable for disease response. The objective response rate (ORR) was 41%, the median progression-free survival (PFS) was 5.0 months, and the median OS was 9.3 months. Additionally, five patients achieved a PR to first-line ALK inhibitor treatment; however, no complete responses (CRs) were registered. Even though there was a small sample size, targeted therapies present novel opportunity [29]. The phase 1 trial is going to assess the side effects and best dose of ceritinib and combination chemotherapy in treating patients with advanced solid tumors (standard chemotherapy plus ceritinib) or locally advanced and metastatic PC (gemcitabine plus nab-paclitaxel plus ceritinib) (NCT02227940). The primary outcomes are the maximum tolerated dose (MTD) and recommended phase 2 dose (RP2D) [30,31].

ALK fusions are rare in PC; however, their identification through genetic profiling can guide targeted therapy and potentially improve the outcomes. Existing ALK inhibitors may achieve substantial and enduring responses, even in patients resistant to previous treatment (Figure 1).

### 2.2. ROS Fusions

The *ROS1* gene encodes a receptor tyrosine kinase (RTK) that is closely related to the *ALK*, and it is expressed in the kidneys, cerebellum, peripheral neural tissue, and GI organs like the stomach, small intestine, and colon. However, the mechanism by which ROS1 fusion proteins become constitutively active remains unknown. It seems that they are involved in common growth and survival pathways that are also activated by other RTKs [32]. Compared to NSCLC (1–2%), ROS1 fusions are less frequent in non-lung solid tumors (≤0.3%), and the fusion partners are mostly unreported [33]. Currently, ROS1 rearrangements in NSCLC are treated with ALK inhibitors—crizotinib and entrectinib—based on the results from the PROFILE 1001 study (NCT00585195) for crizotinib and integrated analysis of the phase 2 STARTRK-2, phase 1 STARTRK-1, and the phase 1 ALKA-372-001 trials for entrectinib [34,35,36,37,38,39].

After disease progression, lorlatinib gives an opportunity for next-line targeted therapy [35,40]. As with ALK fusions, the ROS-1 fusions are rare in PC; however, they may be treated with targeted therapy using ALK inhibitors. One of the case studies showed a durable response to crizotinib and subsequent lorlatinib therapy in a 55-year-old patient with Solute Carrier Family 4 Member 4 (SLC4A4)-ROS1 fusion PDAC [41]. Additionally, a 77-year-old woman was diagnosed with pancreatic acinar cell carcinoma with an ROS1-Centromere Protein W (CENPW) gene. Treatment started with crizotinib twice a day with gemcitabine plus paclitaxel on the first and eighth days. The outcomes were favorable, with decreased CEA levels and sizes of liver metastases, a stabilization of CA19-9 levels, and a shrinkage of the pancreatic mass [42]. Even though in these two cases crizotinib showed clinical activity, another analysis demonstrated that the cohort predominantly showed missense mutations in *ROS1*, which may not be as responsive to ROS1-targeted therapies [43].

Clinical evidence in PC primarily derives from case reports, although it suggests the promise of targeted therapy. Therefore, more attention should be paid to further analyze the efficacy of *ROS1*-targeted therapy.

### 2.3. NTRK Gene Fusions

The *NTRK1*-3 fusions are rare somatic mutations, with a prevalence <1% in solid tumors; however, epidemiological data are limited [44]. In PC, fusions are found in 0.3% of cases, more frequently in WT RAS [45]. They encode tropomyosin receptor kinases A, B, and C (TRKA, TRKB, TRKC), whose expression is mostly limited to the nervous system. The signaling pathway is initiated when neurotrophins bind to TRK receptors, which causes dimerization and the activation of downstream signaling pathways regulating memory, appetite, proprioception, and pain [46].

Based on the latest clinical trials, there are some drugs that may improve the clinical outcomes of the treatment. One of the TRK inhibitors, larotrectinib, showed antitumor activity in TRK fusion-positive cancers in adults and children in NCT02122913, NCT02637687, and NCT02576431 clinical trials [47]. The primary analysis was based on the results from fifty-five patients, where two of them could not be evaluated, 13% had CR, 62% had PR, 13% had stable disease, and 5% had progressive disease. The ORR was 80%. The median duration of response (DOR) and PFS had not been reached. After 1 year, 55% of patients were progression-free. Few adverse events (AEs) of grades 3 or 4 were observed; mostly they were grades 1 or 2. AEs resulting in a dose reduction in eight patients included increased alanine aminotransferase (ALT), aspartate aminotransferase (AST), dizziness, and a decrease in neutrophil count. Even though the results showed clinical activity for larotrectinib, only one patient had PC, so there is a need to further analyze the efficacy of this drug in this specific type of cancer [47]. In the other polled analysis of the previously mentioned clinical trials, there were 159 patients with TRK fusion-positive cancers, and 14 of them had GI malignancies, including 2 pancreatic tumors. At the time of analysis, 102 patients were included, and 24 of them had CR and 97 had PR. Of patients with gastrointestinal tumors, six of them (43%) responded to the treatment; only one had a pancreatic tumor [48].

Another selective TRK, ALK, and ROS1 inhibitor—entrectinib—in the combined results from two phase 1 trials (ALKA-372-001 and STARTRK-1) has shown substantial clinical activity in patients with locally advanced or metastatic solid tumors that have NTRK1/2/3, ALK, or ROS1 fusions who were naïve to prior tyrosine kinase inhibitor (TKI) treatment. Of 119 patients, 60 had confirmed gene rearrangements, and, after excluding those with a prior TKI exposure or subtherapeutic dosing, the so-called “Phase 2-eligible” population comprised 30 patients, of whom 25 were evaluable. In this group, the ORR was 100% in 3 *NTRK1*/*2*/*3*-rearranged tumors, 86% in 14 *ROS1*-rearranged solid tumors, and 57% in 7 *ALK*-rearranged solid tumors [49]. Entrectinib was also assessed in the STARTRK-2 basket study (NCT02568267), with three cases of PC. Two of them had a translocated promoter region (TPR)-NTRK gene fusion, and one had an SCL4-ROS1 gene fusion. Treatment was associated with responses and a prolonged disease control in these patients [50].

To further assess the efficacy of TRK inhibitors in NTRK-positive pancreatic tumors, more attention should be paid to identify the fusion and to a proper analysis of treatment responses in a greater population.

### 2.4. RET Fusions

In PDAC, the RET tyrosine kinase is expressed in 50–65% of cases, particularly in aggressive and high-grade tumors, and is implicated in perineural invasion, associated with poor prognosis [51,52,53]. RET exists as two main isoforms, RET9 and RET51, which differ in signaling potential [54]. An experimental knockdown study showed that RET51 is the predominant driver of invasive behavior, promoting cell motility, polarization, invadopodia formation, extracellular matrix degradation via matrix metalloproteinase 14 (MMP14) induction, and MMP2/9 activation, in a process dependent on tyrosine kinase substrate 5 (TKS5), non-receptor tyrosine kinase SRC, growth factor receptor-bound protein 2 (GRB2), and the GTPase cell division cycle 42 (CDC42) and Ras homolog family member A (RHOA) [52]. RET protooncogene fusion occurs rarely in PC. In one study, only one patient from a group of 160 patients harbored a pathogenic fusion (0.6%) [55]. Tumors (other than NSCLC) harboring this fusion are correlated with worse survival outcomes than in *RET*-WT tumors [56].

There are several RET TKIs like selpercatinib and pralsetinib that have been approved by the U.S. Food and Drug Administration (FDA) for lung and thyroid cancers. The first one was also approved for other tumors harboring RET fusions [57]. The recommendation was supported by results from the LIBRETTO-001 (NCT03157128) trial in patients with RET fusion-positive NSCLC, thyroid cancer, and advanced solid tumors. In this trial, selpercatinib showed a durable efficacy. For previously treated RET fusion-positive NSCLC, *RET*-mutant medullary thyroid cancer, and RET fusion-positive thyroid cancer, the ORR was 64%, 69% (1-year PFS of 82%), and 79% (1-year PFS of 64%). Among previously untreated patients, the ORR was higher—85% for NSCLC and 73% for RET-mutant medullary thyroid cancer (1-year PFS 92%)—while, for RET fusion-positive cancers other than NSCLC and thyroid cancer, the ORR was 43.9%. The most common AEs with grade > 3 were hypertension, increased ALT and AST, hyponatremia, diarrhea, and lymphopenia [57,58,59,60]. In the updated analysis and longer follow-up in RET fusion-positive solid tumors other than lung/thyroid with a focus on GI histology, selpercatinib showed clinical benefits, like CR, PR, and stable disease (SD), in 67.3% of patients. The ORR in a group of patients with pancreatic tumors was 53.8%. No new safety signals were identified [61]. Moreover, the case study of two women with RET fusion-positive PCs showed that the usage of selpercatinib provided sustained disease control [62].

Pralsetinib obtained its approval based on the phase 1/2 ARROW study (NCT03037385) [63,64]. It was further assessed in the prospective analysis of patients with diverse RET fusion-positive tumors, excluding RET fusion-positive NSCLC or thyroid cancer. After the efficacy enrollment cutoff date, twenty-three patients had RET fusion as the only oncogenic driver, and they were available for further evaluation. The ORR was 57%, CR was seen in three patients, and PR was seen in ten patients. Focusing on PC, all four patients showed a noticeable PR, including one CR with a treatment duration of 33.1 months. The median PFS was 7.4 months, and median OS was 13.6 months. However, treatment-related adverse events (TRAEs) occurred in 25 patients (86%), and 20 of them (69%) experienced a grade ≥ 3 TRAE, like increased ALT, AST, neutropenia, and thrombocytopenia. One death also occurred, with an unknown cause [65]. Nevertheless, the results of this study showed a possible therapeutic option for patients with RET fusion-positive solid tumors, even though there is a need for further investigation, especially for PC. One of the case studies also described a remarkable response to pralsetinib in a patient with only the TRIM33-RET fusion, detected in PDAC. A 68-year-old man was treated with oxaliplatin, gemcitabine, and paclitaxel; however, due to a rapid gastrointestinal toxicity and myelosuppression, pralsetinib was administered. After four months, the volume of the primary tumor and metastases decreased by 39% and 39.5%, respectively. PR was confirmed, and PFS was at least 12 months [66].

Due to the more aggressive cancer phenotype, RET fusions in PC require deeper analyses, primarily a surveillance of the response in a larger patient population. There is a need for integration with comprehensive genomic profiling in clinical practice.

### 2.5. FGFR Fusions

The FGFR family mediates cellular responses to fibroblast growth factors, playing key roles in cell proliferation, differentiation, migration, development, and survival. There are four main *FGFR* genes (*FGFR1*, *FGFR2*, *FGFR3*, *FGFR4*) that encode receptors with overlapping but distinct ligand specificities [67]. FGFR signaling is also involved in tumorigenesis by stimulating tumor cell growth, promoting angiogenesis, and developing resistance mechanisms to anticancer therapies [68]. In PC, pathways such as FRS2–MAPK and TGF-β1 induce epithelial–mesenchymal transition (EMT) by activating MMP and degrading the extracellular matrix. FGFR1 and FGFR2 drive the proliferation and differentiation of pancreatic progenitor cells and increase invasiveness. FGFR3 can suppress epithelial tumor growth, and FGFR4 supports cancer cell survival while modulating the genes *ERBB2* and telomerase reverse transcriptase (*TERT*), controlling cell growth and telomerase activity [69].

In other malignancies or in vivo models, FGFR fusions such as FGFR3-TACC and FGFR2–CCDC6 demonstrated oncogenic activity [70,71]. However, PDAC-specific functional validation remains limited. An analysis of 30229 tumors with a clinical diagnosis of PC revealed that FGFR1-*3* alterations such as rearrangements, copy number amplifications, and short variants were present in 6.9% of patients. Known gain-of-function FGFR genomic alterations constitute 1–1.5% of PDAC cases and may confer sensitivity to FGFR inhibitors, leading to durable responses. Additionally, among FGFR rearrangements, *FGFR2* was most frequently affected, with fusions generally occurring in isolation, supporting their role as primary oncogenic drivers [72].

Multiple clinical trials have investigated new agents that would selectively inhibit FGFR signaling, and two of them received FDA approval. The first one—erdafitinib, based on the phase 2 NCT02365597 trial—obtained its approval in the treatment of patients with locally advanced or metastatic *FGFR2*- or *FGFR3*-mutated urothelial carcinoma with progression during or after chemotherapy. This FGFR1-4 inhibitor achieved an objective tumor response in 40% of cases [73]. The results from the phase 2 RAGNAR study (NCT04083976) showed that erdafitinib had robust and clinically meaningful activity in PC patients with prespecified FGFR1-4 fusions. At the data cutoff, the ORR, median DOR, PFS, and OS were 55.6%, 7.1 months, 7.0 months, and 19.7 months, respectively, and responses were seen in cancers with FGFR1 and FGFR2 fusions [74]. Overall, at a median follow-up of 17.9 months, an ORR was observed in 30% patients (64 of 217 patients) across 16 distinct tumor types [75]. The latest EAY131-K1 phase 2 trial showed that, in patients with FGFR1-4 amplification-positive tumors (urothelial carcinoma was excluded), erdafitinib did not meet its primary end point of efficacy as determined by ORR. In the prespecified primary efficacy analysis, five patients of eighteen included reached SD as the best response. The findings demonstrated that FGFR rearrangements and gene mutations—but not amplifications—remain the recognized FGFR alterations with approved indications for FGFR inhibition [76]. The sub-protocol K2 of the phase 2 EAY131 study showed that, in twenty-five patients included in the primary efficacy analysis, the confirmed ORR was 16% of patients meeting the primary endpoint for this study. Moreover, 28% of patients (seven patients) experienced SD. They all had tumors harboring FGFR1-4 mutations or FGFR1-3 fusions. Two patients included in the trial had PDAC; however, no responses to erdafitinib were observed [77].

Although the results from trials are inconclusive due to the small number of patients harboring PCs with FGFR rearrangements, several case studies showed positive outcomes. One of the case studies described a 28-year-old patient with PC and multiple somatic alterations, including FGFR rearrangement in intron seventeen. The patient began treatment with erdafitinib and responded exceptionally, continuing it for over 12 months [78]. Another case study showed that, even in older patients, CR may be observed. A 68-year-old woman with stage 4 pancreatic carcinoma harbored FGFR2-TACC2 fusion and received erdafitinib. A durable CR was maintained for 10 months; however, grade 2 hyperphosphatemia, diarrhea, and transient central serous retinopathy appeared [79].

The second FGFR inhibitor—pemigatinib—was approved for patients with pretreated locally advanced unresectable or metastatic cholangiocarcinoma with FGFR2 fusion or rearrangement based on the phase 2 FIGHT-202 trial (NCT02924376). This FGFR1-3 inhibitor had an ORR of 35.5%, which supports the therapeutic potential of pemigatinib [80]. Its efficacy was also assessed in the phase 2 FIGHT-207 basket study (NCT03822117) of FGFR1-FGFR3-altered advanced solid tumors. Patients were divided into three cohorts: A, with fusions and rearrangement, B, with activating non-kinase domain single nucleotide variants, and C, with kinase domain mutations or variants of unknown significance. Among these cohorts, the ORR was 27%, 9.4%, and 3.8%, respectively. The DOR, PFS, and OS were longer in cohort A in comparison to cohort B. Moreover, in cohort A, among forty-nine patients, eight of them had PC, and pemigatinib demonstrated its activity. However, one patient harboring FGFR1-PDE4DIP fusion developed clinical on-target resistance to an FGFR inhibitor driven by mutations occurring near the gatekeeper and molecular brake residues [81]. In the phase 2 telemedicine trial, pemigatinib is going to be assessed in FGFR-altered pancreatic cancer, where the primary cohort includes patients with FGFR2 fusions and the secondary cohort includes patients with other known activating mutations in *FGFR.* The primary objective is ORR and is expected to reach ≥33% and <10% as an unfavorable response [82].

The current inhibitors continue to target FGFR fusions as a therapeutic approach, despite occasional unfavorable outcomes. More patients with advanced PC and FGFR2 fusions were treated with FGFR inhibitors, and they had durable responses and prolonged survival [72,83]. Acquired resistance remains a challenge, and it should be further investigated.

### 2.6. NRG Fusions

The Erythroblastic Leukemia Viral Oncogene (ERBB) family consists of four members: ERBB1/HER1, ERBB2/Neu/HER2, ERBB3/HER3, and ERBB4/HER4 [84]. The *NRG1* gene encodes ligands for the ERBB family, which promotes the heterodimerization of ERBB2 and ERBB3. Then, the ERBB2/ERBB3 heterodimer activates WT RAS, PI3K–AKT, and MAPK signaling pathways, promoting cellular proliferation and the evasion of apoptosis [85,86]. *ERBB3* is not often mutated or amplified in cancer; however, the overproduction of ligands is another way that malignancies abnormally activate ERBB receptors, such as through the presence of an autocrine loop between ERBB3 and NRG1 [85,87].

*NRG1* fusions are rare, with a prevalence of <1%; the blockage of this loop could be an opportunity for patients harboring *NRG1*-rearranged tumors [88,89,90]. As part of a prospective clinical trial, after whole-genome sequencing (WGS), 3 patients from 47 had WT KRAS tumors with fusions *NRG1*-ATPase Na^+^/K^+^-transporting subunit beta 1 (*ATP1B1*) and *NRG1*-Amyloid beta precursor protein (*APP*) [91]. Other fusion partners that could be identified are cluster of differentiation 74 (*CD74*), cadherin 6 (*CDH6*), Store-operated calcium entry-associated regulatory factor (*SARAF*), and Rho-associated coiled-coil-containing protein kinase 1 (*ROCK1*) genes [91,92,93].

Pan-ERBB TKIs offer a therapeutic approach in the treatment of *NGR1*-rearranged solid tumors. A retrospective non-comparative cohort study assessed afatinib, a second-generation irreversible pan-EGFR family kinase inhibitor, for *NRG1*-rearranged solid tumors. The drug produced a modest overall activity, with CR and PR mainly in NSCLC. The ORR was 37.5% overall and 43.8% when received as a first-line therapy. Among eleven patients with PDAC, the ORR was 9.1%, with one PR, and the median PFS and OS were 4.6 months and 6.9 months, respectively. Toxicity was manageable and consistent with well-characterized AE profiles for afatinib. The non-afatinib group also had high response rates. In six PDAC patients, the ORR was 66.7%, and four of them had PR, so prospective clinical trials in patients with *NRG1* fusion-driven PDAC are needed to evaluate the efficacy of different treatments, including afatinib [94].

In a case series, afatinib showed activity in patients with metastatic colorectal and lung cancers. Several patients demonstrated a prolonged response of more than 18 months [95]. It also showed activity toward a subset of PC cells closely correlated with the expression of EGFR, human epidermal growth factor receptor 2 (HER2), and HER3 receptors [96]. One patient with *ATP1B1–NRG1* fusion-positive metastatic PDAC, after progression on FOLFIRINOX treatment, started monotherapy with afatinib, which resulted in a substantial decrease in the size of liver metastases after 7 weeks. However, progression was seen 3 months later [92].

ERBB-targeted therapies using monoclonal antibodies could provide clinical improvements in survival outcomes. Firstly, zenocutuzumab (MCLA-128)—a bispecific antibody against HER2 and HER3—in in vitro studies demonstrated potent antitumor activity in cell lines harboring NRG1 fusions, leading to the suppression of proliferation and induction of cell death. Moreover, two patients with *ATP1B1–NRG1*-positive PC achieved rapid responses with treatment durations of over 12 months. Another patient with *CD74–CD74-NRG1*-positive NSCLC had progressed in six prior lines of systemic therapy, including afatinib; however, with zenocutuzumab, the patient responded rapidly to treatment with a PR [97]. The phase 2 eNRGy study (NCT02912949) showed a 30% response in patients with twelve tumor types. In patients with PC, 15 of 36 patients (42%) responded. The median PFS was 6.8 months, and the most common mainly low-grade AEs were diarrhea, fatigue, and nausea [98]. Based on the trial results, zenocutuzumab obtained FDA approval for adults with advanced, unresectable, or metastatic NSCLC and PDAC harboring *NRG1* gene fusion with disease progression during or after prior systemic therapy [99].

Secondly, seribantumab—a monoclonal antibody that blocks HER3 activation by NRG1—led to the inhibition of *NRG1* fusion-dependent tumor growth in patient-derived breast, lung, and ovarian cancer models [100]. The phase 2 CRESTONE study (NCT04383210) showed durable responses in advanced solid tumors harboring *NRG1* fusions and had a favorable safety profile with low-grade AEs: diarrhea, fatigue, and rash. In cohort 1 (patients with solid tumors harboring *NRG1* fusions who received at least one prior therapy and were naïve to ERBB-targeted therapy), among 10 patients, the confirmed ORR was 30%, and the disease control rate (DCR) was 90% [101]. Updated efficacy data from cohort 1 indicates that seribantumab has an acceptable safety, tolerability profile, and durable clinical activity. Among 22 patients, the ORR was 36% and DCR was 95% [102]. The results showed that HER3 blockade is a rational approach.

Combining an ERBB2 monoclonal antibody—pertuzumab—and an EGFR tyrosine kinase inhibitor—erlotinib—showed activity in relapsed NSCLC; however, it was poorly tolerated, and 68.3% of patients experienced TRAEs grade ≥3 [103]. One patient with *NGR1*-rearranged PDAC received third-line therapy consisting of gemcitabine and erlotinib, which, due to progression, was switched to erlotinib and pertuzumab. After 8 weeks, a partial remission of liver metastases and normalization of serum CA19-9 levels were observed; nevertheless, the disease progressed after 3 months [92].

Further research is needed to fully assess the treatment possibilities of *NRG1* gene fusions in WT KRAS PCs. To date, zenocutuzumab has demonstrated the most promising clinical activity among agents against *NRG1*. Additional ERBB3-targeting therapies or combination strategies that target the ERBB–RAS–MAPK signaling pathway should be further taken under investigation (Table 1) [91].

## 3. Gene Amplifications in Pancreatic Cancer

HER2 (ERBB2) amplification occurs with a 2.2% prevalence in WT PDAC, which makes it another therapeutic target [104]. The trastuzumab plus pertuzumab combination is a therapeutic option for patients harboring HER2 amplification. Data from two cohorts of patients with colorectal cancer with either *ERBB2* amplifications or *ERBB2* or *ERBB3* mutations showed that treatment with pertuzumab plus trastuzumab had antitumor activity only in *ERBB2* amplification. The ORR was 25% compared with 0%, respectively [105]. In the phase 2a MyPathway trial (NCT02091141) among nine patients who had PC with *ERBB2* amplification or overexpression and were treated with trastuzumab plus pertuzumab, only two patients had PR, and one patient had SD >120 days [106]. A further evaluation of the effectiveness of PC treatment with pertuzumab plus trastuzumab is needed.

The phase 3 KEYNOTE-811 study (NCT03615326) assessing pembrolizumab plus trastuzumab and chemotherapy for patients with HER2-positive advanced, unresectable, or metastatic gastric and gastroesophageal junction (G/GEJ) adenocarcinoma showed that this combination provided a statistically significant and clinically meaningful improvement in OS [107]. Therefore, using combination therapy with anti-HER2 therapy may be an option to improve clinical outcomes in patients with HER2-altered PDAC. One patient with widely metastatic PDAC, with high-level HER2 amplification, received multimodal therapy (anti-HER2 antibody–drug conjugates (ADC) trastuzumab-deruxtecan (T-DXd), checkpoint blockade, stereotactic radiotherapy, personalized neoantigen vaccine). This combination therapy led to a rapid and complete remission, sustained over follow-up, underscoring the potential benefit of integrating genomic testing, functional organoid assays, and targeted therapies in selected PC cases [108].

Other rare gene amplifications reported in WT KRAS PDAC include FGF3 (3%), FGFR3 (1.8%), NTRK (1.8%), and MET (1.3%) [104]. While FGF3 amplification often co-occurs with other alterations like CCND1, FGF4, and FGF19 [109], there is a lack of clinical evidence on treatment possibilities in PDAC. NTRK amplification, unlike NTRK fusions, has not been associated with meaningful responses to TRK inhibitors. In one exception, 53 patients whose tumors had point mutations, amplifications, copy number variations, insertions, or deletions involving NTRK, ROS1, or ALK did not exhibit any objective responses to entrectinib in the ALKA 372–001 and STARTRK-1 studies [49,110]. For patients with advanced or metastatic NSCLC, gastric cancer, or solid tumors harboring genetic alterations in MET, the SHIELD-1 trial (NCT03993873) is going to assess the safety, tolerability, PK, and preliminary efficacy of elzovantinib (TPX-0022) [111]. Given the low prevalence and the absence of PDAC-specific prospective studies, these alterations currently represent potential, yet unvalidated, therapeutic targets, highlighting a critical area for future research.

## 4. Gene Mutations in PDAC

### 4.1. BRAF

About 13% of WT KRAS PDAC had *BRAF* mutations, which was, after *TP53*, the most frequent mutation [104]. Those mutations can be divided into three classes, based on the mechanism of action:•The first one includes the most frequent V600E mutation, which can be targeted by BRAF inhibitors, optionally combined with MEK inhibitors.•The second class consists of in-frame deletions that can be targeted by MEK inhibitors.•The third class can be targeted by MEK inhibitors with RTK inhibitors [112].

KRAS-WT PDAC with V600E mutation has an FDA-approved combined therapy with dabrafenib (BRAF inhibitor) and trametinib (MEK inhibitor) [112]. Inhibitory synergism was shown in preclinical studies because this combination of MEK and BRAF inhibitors overcomes the paradoxical MAPK activation (induced by BRAF inhibitors, which are suspected to be a reason for therapeutic resistance) in PDAC in vitro [113]. Another combination of these mechanisms was used in the NCT04390243 trial with encorafenib and binimetinib and described in a case study in which the drug combination was more effective, as evidenced by the disease progression following the discontinuation of binimetinib and the PR upon its reintroduction [114]. A retrospective study by Ben-Ammar et al. observed that this combination of targeted therapy improves both OS and PFS for patients with WT KRAS PDAC [18]. However, in cases of *BRAF* fusion, a study has shown no benefits of regorafenib (dual-targeted Vascular Endothelial Growth Factor Receptor 2-Tyrosine Kinase with Immunoglobulin-like and EGF-like Domains 2 (VEGFR2-TIE2) inhibitor) and trametinib in PDAC patients [115]. The MATCH trial will assess the effectiveness of ulicertinib in patients with *BRAF* fusions [116].

### 4.2. HER2

In a study by Singhi et al. using a targeted genomic profile analysis of a large number of PDACs, it was estimated that HER2 mutations consist of 3% WT KRAS cases [117]. Trastuzumab could potentially be beneficial in targeting HER2 mutations in PDAC, but unfortunately clinical trials have not had satisfactory results in improving PDAC prognosis [118]. The trial NCT00923299 combined trastuzumab with cetuximab but faced severe toxicity issues, mainly skin-related [119]. Another DESTINY-PanTumor02 (NCT04482309) trial combined T-DXd and concluded that therapy is beneficial for HER2-expressing solid tumors, with especially good results observed in patients with IHC 3+ tumors [120]. Another multicenter phase 2 trial study showed that the combination of trastuzumab and capecitabine in patients with HER2-overexpressing metastatic PC did not improve PFS and OS compared with standard chemotherapy [121]. A different approach in treatment was evaluated in a randomized, open-label phase 2 ACCEPT study of the Arbeitsgemeinschaft Internistische Onkologie. A comparison of afatinib plus gemcitabine versus gemcitabine alone as a first-line treatment was performed, and it unfortunately concluded that adding afatinib to gemcitabine did not improve therapeutic efficacy and resulted in more AEs, for example, diarrhea (71% vs. 13%) and rash (65% vs. 5%) [122].

### 4.3. PIK3CA

In a recent study by Pitiyarachchi et al. in a group of 6640 patients with PDAC, *PIK3CA* somatic mutations (*E545K*, *E542K*, *H1047R*) were found in 2.7%, with a similar distribution across age, gender, and race, but—what is worth mentioning—with a slightly higher prevalence in metastatic samples. They are, in most cases, recognized as oncogenic and have shown predictive responses to PIK3CA inhibition targeting the PI3K/AKT/mTOR pathway [123]. Furthermore, PI3K/mTOR dual inhibitors are being assessed in vitro and in vivo with promising results for PDAC patients [124]. A phase 2 study had promising results, including 31 patients with advanced PDAC receiving a combination of everolimus and capecitabine, which has been proven safe and non-toxic, and resulted in an OS rate of 8.9 months [125]. Another therapeutic option that was considered was metformin—an antidiabetic drug that can inhibit the mTOR pathway by activating AMP-activated protein kinase (AMPK), but it was disproven that metformin could enhance the antitumor effect of gemcitabine and erlotinib in a large, double-blind, randomized, placebo-controlled phase 2 trial [126]. On the other hand, a smaller study examined metformin with or without rapamycin as maintenance therapy after induction chemotherapy in patients with metastatic PDAC, and this treatment was well tolerated and several patients achieved stable disease associated with prolonged survival, so the authors suggested a need for further prospective research [127]. Other drugs currently in clinical trials targeting PI3K/mTOR are, for example, dactolisib, voxtalisib, omipalisib, and gedatolisib [124,128,129,130,131].

### 4.4. PTEN R130Q/STK11/TSC2

A study in a mouse model showed that the conditional inactivation of phosphatase and tensin homolog deleted on chromosome 10 (PTEN), a negative regulator of the PI3-K/AKT signaling pathway, led to metaplastic PDAC development. Loss-of-function mutations of PTEN are seen in several human solid cancers, including PC [132]. PTEN-deficient tumors additionally have a strong activation of the Nuclear Factor kappa B (NF-κB)-cytokine network, which provides additional avenues for targeted therapies in tumors with altered PI3K regulation [133]. STK11, a classic tumor suppressor, is mutated in PDAC in 4% of sporadic PDAC [134,135]. A recent study’s findings support that a phosphodiesterase inhibitor, roflumilast, has a target profile that makes it a potential agent for therapeutic intervention for STK11 mutant PC patients [135].

## 5. Chromatin Remodeling Genomic Alteration

### 5.1. ARID1A

Mutations in genes involved in chromatin remodeling, like AT-Rich Interaction Domain 1A/1B (*ARID1A*/*ARID1B*), Polybromo 1 (*PBRM*), AT-Rich Interaction Domain 2 (*ARID2*), Lysine Methyltransferase 2D (*KMT2D*), Lysine Methyltransferase 2C (*KMT2C*), SWI/SNF-Related, Matrix-Associated, Actin-Dependent Regulator of Chromatin, Subfamily A, Member 4 (*SMARCA4*), and SET Domain-Containing 2 (*SETD2*), are not common in WT KRAS PC except for one gene. *ARID1A* gene mutations occur with an 11.6% prevalence, based on the results from a retrospective study [104]. It encodes ARID1A, ARID1B, and ARID2—a switch/sucrose non-fermentable (SWI/SNF) chromatin remodeling complex subunit responsible for binding DNA. ARID1A plays a pivotal role in modulating gene expression programs that can either promote tumorigenesis or suppress it. Functionally, it influences multiple cancer-relevant pathways, including PI3K/AKT/mTOR signaling, immune evasion mechanisms, mismatch repair regulation, enhancer of zeste homolog 2 (EZH2)-mediated methylation, steroid receptor signaling, DNA damage checkpoints, p53 target regulation, and KRAS-driven transcriptional networks [136].

Several targets for ARID1A-deficient PDAC are being investigated, including poly ADP ribose polymerase (PARP), EZH2 (NCT05023655), and the PI3K/Akt/mTOR pathway (Figure 2) [137,138,139]. Additionally, the immune checkpoint blockade (ICB) is being evaluated in clinical trials. In Okamura R et al.’s study (2020), the median PFS after ICB was significantly longer in the patients with *ARID1A*-altered tumors than in those with *ARID1A* WT tumors (11 months vs. 4 months, *p* = 0.006) [140]. In Botta et al.’s study (2021, NCT02478931), nine patients with metastatic PDAC harboring SWI/SNF chromatin remodeling gene alterations received ICB. The ORR was 89%, including a CR, and the two longest responses are ongoing at more than 33 and more than 36 months. The median PFS and OS were 9 months and 15 months, respectively. The results showed that, even in a highly immunosuppressive microenvironment, immunotherapy in SWI/SNF-altered metastatic PC patients has clinical relevance and showed good response [141]. Further trials should explore the ICB in a greater population.

### 5.2. SMARCA4

SMARCA4, also known as Transcription Activator BRG1, is an ATPase not only involved in chromatin remodeling but also in DNA damage repair [142]. A rare case of SMARCA4-deficient undifferentiated PC with a very low survival rate was recently reported [143]. In NSCLC, an aberrant expression of SMARCA4 occurs in approximately 10% of patients [144], and it is associated with a weak response to conventional chemotherapy and poor prognosis. However, there is a high possibility that cisplatin-based chemotherapy, in combination with immune checkpoint inhibitor (ICI) therapy, ataxia telangiectasia and Rad3-related (ATR) kinase inhibitors (berzosertib, emilisertib), CDK4/6 inhibitors (palbociclib, ribociclib, abemaciclib, trilaciclib), OXPHOS inhibitors, EZH2 inhibitors (tazemetostat), and Aurora Kinase A Inhibitors (VIC-1911, alisertib, LY3295668), has potential as a therapeutic option for *SMARCA4*-deficient tumors [145].

In SMARCA4-mutant cancer cells, SMARCA2 becomes essential for survival (synthetic lethality). Targeting SMARCA2—using Proteolysis-Targeting Chimera (PROTAC) degraders such as PRT3789, YD23, and A947—may selectively impair the growth of SMARCA4-deficient PDAC cells while sparing normal tissue [146,147,148]. PRT3789 is currently under evaluation in phase 1 and phase 2 studies in patients with advanced solid tumors with a loss of SMARCA4 (NCT05639751 and NCT06682806) [148]. In the phase 1 NCT05639751 study, it showed excellent pharmacodynamic effects with signs of antitumor activity. The most common AEs reported were nausea, constipation and dyspnea, decreased appetite, fatigue, and anemia [149]. Recently, SMD-3236—a SMARCA2 degrader—has been identified, which potently inhibited cell growth in SMARCA4-deficient cell lines and displayed a minimal activity in SMARCA4 WT cell lines. It was also well-tolerated in the xenograft model [150].

Innovative therapeutic approaches targeting the SWI/SNF complex may therefore represent a promising opportunity for this difficult-to-treat patient population.

## 6. Mismatch Repair Deficiency and Microsatellite Instability

Mismatch repair (MMR) genes encode proteins (MutS and MutL homologues) responsible for the strand-specific correction of mispaired bases by removing base–base and small insertion–deletion mismatches [151]. In humans, in the MutS group, there are MSH2, MSH3, and MSH6, and the MutL group includes MLH1, MLH3, and post-meiotic segregation (PMS1, PMS2) nuclear factors [152]. Additionally, the MMR system relies mainly on creating MSH2-MSH6 (MutSalpha) and MSH2-MSH3 (MutSbeta) heterodimers, which recognize replication-associated errors. Once bound, they recruit MutL homolog complexes, primarily MLH1-PMS2 (MutLalpha) with ATPase activity, that coordinate the excision of the faulty DNA segment and its resynthesis [153,154,155,156]. Beyond correcting replication errors, MMR also prevents aberrant recombination events and safeguards genomic stability [157].

Intraductal Papillary Mucinous Neoplasm (IPMN) is connected with microsatellite instability (MSI)/MMR deficient (dMMR) in 6.9% of cases, whereas dMMR in PDAC is rare—1–2%—and strongly connected with a *KRAS*/*TP53* wild-type molecular background [158,159]. Additionally, young-onset PC (in patients younger than 50 years old) shows higher proportions of MSI-high/dMMR, *BRCA2*-mutant, and *PALB2*-mutant tumors compared with patients with average-onset pancreatic cancer (70 years and older) [160].

Immunotherapy has not shown a significant efficacy in pancreatic cancer due to its immunosuppressive microenvironment [161,162]. However, MSI-high/dMMR tumors demonstrate increased immunogenicity and immune checkpoint proteins such as programmed death-1 and -ligand 1 (PD-1 and PD-L1) overexpression, suggesting potential benefits from ICB [162,163,164,165].

The anti-PD-1 monoclonal antibody, pembrolizumab, showed clinical activity in MSI-high/dMMR colorectal and noncolorectal cancers [166]. In the phase 2 KEYNOTE-158 trial (NCT02628067) among 233 patients where the most common cancers were endometrial, gastric, cholangiocarcinoma, and pancreatic, the ORR, median PFS, and median OS were 34.3%, 4.1 months, and 23.5 months, respectively. In the pancreatic cancer (22 patients) subgroup, the ORR, median PFS, and median OS were 18.2%, 2.1 months, and 4 months, respectively. One patient achieved CR, and three achieved PRs. The most common TRAEs were fatigue, pruritus, diarrhea, and asthenia, and the most common immune-mediated AEs were hypothyroidism, hyperthyroidism, colitis, and pneumonitis [167]. The results were consistent with the previous results of the KEYNOTE-016 trial (ORR 53%) [168]. Based on the analysis of five single-arm studies—including KEYNOTE-158, -016, -164, -012, and -028—pembrolizumab received FDA approval for unresectable or metastatic MSI-high/dMMR solid tumors that have progressed after prior standard treatment and have no satisfactory alternative treatment options, or with MSI-high/dMMR colorectal cancer that has progressed after treatment with a fluoropyrimidine, oxaliplatin, and irinotecan [169].

Dorstalimab, an IgG4 anti-PD-1 monoclonal antibody, demonstrated durable responses in dMMR tumors, with a 100% remission rate in a phase 2 trial of dMMR rectal cancer (NCT04165772) [170]. In the phase 1 GARNET trial (NCT02715284), it yielded an ORR of 44% and median PFS of 6.9 months in MSI-high/dMMR solid tumors. Moreover, 72.2% of responders had a response lasting 12 or more months. In the pancreatic cancer group (12 patients), 5 patients achieved PR, and the ORR was 41.7% [171]. Dorstalimab received FDA approval in 2021 for dMMR recurrent or advanced solid tumors that have progressed during or following prior treatment and that have no satisfactory alternative treatment options [172]. Combination approaches were evaluated in the phase 1b IOLite trial (NCT03307785). A triplet combination of dostarlimab plus niraparib (PARP inhibitor) plus bevacizumab (VEGF inhibitor) showed the highest disease control, though only three pancreatic cancer patients were included. It was well tolerated, with grade ≥ 3 treatment-emergent adverse events (TEAEs) across all parts: anemia, thrombocytopenia, and neutropenia [173].

Tumors classified as homologous recombination deficiency (HRD) cancers, but with a predominant dMMR signature, may have an elevated mutation burden and be candidates for treatment with ICIs; conversely, a perturbed HR system in classically MSI-affected tumors may represent an actionable biomarker for treatment with PARP inhibitors [174]. There is also the possibility of combining these inhibitors, resulting in synergistic tumor control. In the phase 1b/2 study of niraparib plus anti-PD-1 (nivolumab) or anti-CTLA-4 (ipilimumab), the combination of niraparib plus ipilimumab met the primary endpoint of a superior PFS rate at 6 months (59.6%) and 17.3 OS in patients with advanced PC whose cancer had not progressed after ≥16 weeks of platinum-based therapy. Moreover, the effect was independent of a clinically identified DDR deficiency. However, 50% of patients in the niraparib plus ipilimumab group experienced a grade 3–4 TRAE [175]. Further studies: POLAR (NCT04666740), NCT04493060, and NCT04548752 are exploring combinations of PARP inhibition plus anti-PD-1. Compared to patients with microsatellite-stable (MSS), MMR-proficient, non-Lynch syndrome-associated PDACs, patients with MSI-high, dMMR, and Lynch syndrome-associated PDACs showed a considerably higher survival rate prior to the routine use of immune checkpoint medications [176]. That is why MSI should be assessed, using NGS for analyzing all potential therapeutic targets and even better survival outcomes [159].

## 7. Germline Mutations in PDAC

Germline DNA damage repair pathway (DDR) mutations in WT KRAS PDAC are not directly related to KRAS mutations but rather represent a separate genetic pathway. Understanding the presence of these mutations is crucial for personalized treatment strategies and potentially improving patient outcomes in WT KRAS PDAC. Germline variants can be passed from parents to offspring and are well-known molecular alterations in cancer-predisposing genes. Approximately 4.7% of PDAC patients carry the *BRCA1* and the *BRCA2* gene mutations, and 2.5% harbor HRD-related genes, such as *ATM*, checkpoint kinase 2 (*CHEK2*), and *PALB2* [177]. In high-risk groups, including familial pancreatic cancer (FPC), up to 30% of patients carry germline pathogenic variants [178]. The majority of linked genes belong to DNA damage repair pathways, specifically for mismatch repair and homologous recombination repair (HRR), such as *BRCA1/2*, *ATM,* and *PALB2* [179,180]. The Cancer Genome Atlas Research Network indicates that germline mutations in familial predisposition genes are significantly enriched in WT KRAS PDAC (*p* = 0.027), particularly in *ATM* or *PRSS1* (causing familial pancreatitis) [181]. Other analysis identified multiple mutations, especially in *BRCA2*, *ATM*, *BAP1*, *RAD50*, *FANCE*, and *PALB2* genes [104].

Regardless of ethnicity or family history, all patients with PDAC should undergo germline genetic testing, according to the current National Comprehensive Cancer Network (NCCN) guidelines [182]. When more than one relevant gene is involved in a malignancy, including PDAC, the American Society of Clinical Oncology (ASCO) also supports the use of germline multi-gene panel testing [183]. To differentiate pathogenic or possibly pathogenic variants from variants of uncertain significance (VUS), germline assessment is usually performed using NGS multigene panels with deletion/duplication analysis, followed by variant interpretation based on ACMG/Association for Molecular Pathology (AMP) criteria [184,185,186].

### 7.1. BRCA1/2

Tumor suppressor genes *BRCA1* and *BRCA2* are essential for preserving genomic stability. They are key components of the HRR mechanism that repairs DNA double-strand breaks (DSBs) [187]. In about 3–7% of patients with PDAC, germline mutations are found in the *BRCA1* and *BRCA2* genes, making them among the most frequent hereditary alterations in PDAC [185,188,189]. Some studies, such as this one performed by Yurgelun et al., report the frequency to be up to 8% in unselected patients undergoing germline testing. Moreover, *BRCA2* mutations occur more frequently than *BRCA1* mutations, accounting for roughly two-thirds of BRCA-related PDAC. Only about 0.2% of PDAC patients had pathogenic *BRCA1* mutations, according to population-based studies like the UK Biobank and Geisinger cohorts, indicating a low background frequency in the general population [190]. Compared to sporadic instances, FPC kindreds, defined as having two or more first-degree relatives with PDAC, have significantly higher levels of *BRCA1/2* mutation rates, up to 20% in FPC cohorts [184].

Loss-of-function germline mutations in *BRCA1* or *BRCA2* damage the HRR pathway, leading to HRD and increased genomic instability in PDAC cells. As a result, there is an increase in DNA double-strand breaks, chromosomal rearrangements, and a mutator phenotype, which encourages the proliferation of the malignancy [191,192]. A loss of heterozygosity (LOH) and other tumors with *BRCA1/2* biallelic inactivation have the greatest HRD metrics and genomic scarring markers [191,193]. Compared to non-mutant cases, *BRCA1/2*-mutant PDAC has an increased susceptibility to platinum-based chemotherapy, leading to a greater response rate and longer PFS [193,194,195]. In metastatic PDAC patients with genetic *BRCA1/2* mutations whose disease remained stable following first-line platinum therapy, the POLO study showed that maintenance therapy with the PARP inhibitor olaparib significantly extended PFS, leading to its approval as a maintenance treatment for this patient subgroup [196]. In a study performed by Boursi et al., *BRCA2* mutation carriers showed a significantly improved OS compared to *BRCA1* carriers (29 vs. 23 months; 56 vs. 24 months after resection and use of platinum drugs) [197]. Because of mechanisms such as secondary mutations, replication fork stabilization, and reversion mutations, many *BRCA*-mutant PDAC patients become platinum- or PARP inhibitor-resistant despite their initial high sensitivity, which is a leading problem in the long-term management of *BRCA*-mutant PDAC [192,193,195].

According to current clinical guidelines, all PDAC patients should undergo germline genetic testing for *BRCA1/2* and other key HRR genes to determine eligibility for PARP inhibitor therapy [198,199]. In addition to germline testing, RAD51 foci formation assays and immunofluorescence on tumor tissue following DNA damage are becoming more validated as useful indicators of homologous recombination proficiency. When compared to single-gene mutation detection or genomic scar metrics, these studies have shown a higher predictive accuracy for responses to platinum-based chemotherapy and PARP inhibitors. Combining germline BRCA testing with RAD51 functional assays helps define the HRD phenotype and improves patient selection for targeted therapy. In order to improve diagnostic accuracy in PDAC, next-generation HRD testing techniques now use genomic scar signatures, mutational signature algorithms, and machine learning classifiers that combine with RAD51 assay results [200,201,202,203].

Platinum-based regiments (e.g., FOLFIRINOX) remain the first-line treatment for *BRCA1/2*-mutated PDAC, exploiting the inability of HR-deficient tumors to repair DSBs. It is speculated that a prolonged exposure to platinum may result in selection pressure and the end development of resistant subclones; therefore, early intervention when tumors remain HRD-positive is recommended [204,205]. Ongoing trials combine PARP inhibitors with ataxia telangiectasia and RAD3-related kinase inhibitors (ATR inhibitors), PI3K inhibitors, or ICB to repair DNA damage signaling or induce immunogenic cell death in HRD tumors. These strategies may be particularly helpful for patients with partial *BRCA* reversion or restored RAD51 function [205,206].

### 7.2. PALB2 (HRD)

PALB2 functions as a scaffold protein linking BRCA1, BRCA2, and RAD51 to facilitate HRR at DSB sites [207,208]. Next, through a mechanism functionally analogous to BRCA1/2, it directly recruits BRCA2 and RAD51 to damage DNA and facilitates the production of RAD51 filaments and strand invasion [209,210,211]. To strengthen genomic integrity, PALB2 additionally stabilizes RAD51 filaments, even when the replication protein is present, increases RAD51-mediated D-loop formation, and binds DNA via its DNA-binding domain. PALB2, just like BRCA1/2, acts as a tumor suppressor, and pathogenic germline mutations increase the risk of breast, ovarian, and pancreatic cancers, underscoring its shared involvement in DNA repair [207,210,212,213].

*PALB2* mutation in PDAC is relatively rare, typically occurring in 0.5% to 1.5% of unselected PDAC cohorts; some studies, such as a large Chinese population analysis, report a 1.8% prevalence [214,215]. It should also be stated that the frequency was still lower than that of *BRCA1* and *ATM*. Overall, *PALB2* plays a minor role in the total burden of HRD in comparison to more dominant players like *BRCA2* and *ATM* [216,217].

Like *BRCA1/2*, germline *PALB2* mutations cause HRD and genomic instability in tumors, which enable DNA damage and may affect the aggressiveness of the tumor [210]. It has been observed that patients with *PALB2*-mutated PDAC often show a greater sensitivity to DNA-damaging therapies, such as platinum-based chemotherapy and PARP inhibitors. Moreover, *PALB2* mutation carriers, frequently grouped with *BRCA2* carriers, exhibited an improved therapeutic responsiveness and possibly extrapolated survival benefits, particularly when tumors are HRD-positive [194,205].

In patients with PDAC or a family history of breast and ovarian cancers, germline *PALB2* testing is performed using multigene panels. *PALB2* germline mutations are inherited in an autosomal dominant manner, and, when a pathogenic mutation is found in first-degree relatives, cascade testing is advised for determining the cancer risk [218]. According to the American College of Medical Genetics and Genomics (ACMG) guidelines, VUS should not be used to guide clinical management, due to the fact that only a small portion of the >600 known *PALB2* variants are pathogenic [208].

As with BRCA-mutated tumors, *PALB2* mutations show an increased sensitivity to platinum-based chemotherapy, demonstrating superior response rates and a prolonged real-world PFS compared to non-HRD tumors. In tumors with *PALB2* mutations, PARP inhibitors have demonstrated promise by using synthetic lethality to take advantage of the homologous recombination deficit. Rucaparib, used as a maintenance therapy, has shown multi-year progression-free remission in *PALB2*-mutated PDAC. Platinum-based combinations are considered standard care; however, emerging treatment frameworks now include gemcitabine-cisplatin [194,205,219,220,221].

### 7.3. ATM

As a serine/threonine protein kinase essential to the DNA damage response, ATM phosphorylates key substrates (including p53, *BRCA1*, and *H2AX*) in response to DNA DSBs, hence coordinating repair, cell cycle arrest, or apoptosis [222,223]. *ATM* deletion in PDAC models highlights its tumor-suppressive function in pancreas-specific oncogenesis by accelerating tumor growth, promoting EMT, increasing mitotic abnormalities, and causing genomic instability [224]. In unselected PDAC patients, approximately 2–3% carry an *ATM* germline pathogenic mutation, which is highly associated with an elevated PC risk. As a result, *ATM* pathogenic variants are recognized as susceptibility markers [225]. A therapeutic vulnerability is created by *ATM* dysfunction: ATR or PARP inhibitors may be used to specifically target *ATM*-deficient tumors, due to the fact that PDAC cells without functional *ATM* become more dependent on ATR- and checkpoint kinase 1 (CHEK1)-mediated repair pathways [226].

In pancreatic models, the loss of *ATM* function suggests a more aggressive tumor phenotype by promoting genomic instability, EMT, and quicker tumor growth [227]. Moreover, clinical tissue studies suggest that the loss of (phospho)-ATM expression correlates with a poorer prognosis and lower gemcitabine sensitivity, supporting a correlation between *ATM* pathway failure and unfavorable behavior in PDAC [228]. In resected human PDAC, tumoral *ATM* loss, especially with intact TP53, was independently related to a lower OS, suggesting that *ATM* loss indicates a high-risk subpopulation [229]. Conversely, a multicenter cohort of patients with pathogenic *ATM* mutations revealed equal or even superior outcomes against comparators, underlining heterogeneity that may reflect differences in zygosity (germline vs. biallelic loss), co-drivers, for example, TP53 status, and medication taken [230]. Overall, pathological and experimental evidence links *ATM* deficiency to a more invasive biology; nevertheless, clinical survival statistics are inconsistent and probably rely on the co-mutation context (e.g., TP53), therapeutic exposure, and whether tumors achieve biallelic *ATM* inactivation [224,226,231].

When a germline *ATM* mutation is discovered, several centers use paired tumor-normal sequencing to document biallelic inactivation because the clinical impact frequently depends on whether tumors acquire a second hit (such as a loss of heterozygosity or a somatic *ATM* change) [230]. Although germline testing is necessary to differentiate inherited from merely somatic loss, ATM protein loss by immunohistochemistry (IHC) in resected PDAC is associated with poor behavior and serves as a tissue-level indicator of pathway disturbance [228,229]. Based on the International Cancer of the Pancreas Screening Consortium (CAPS) consensus criteria, carriers with a family history of PC may be eligible for high-risk surveillance (such as Magnetic Resonance Imaging (MRI) or EUS if a pathogenic germline *ATM* variation is found. It is also advised to perform a cascade testing of at-risk relatives [184,232].

Since *ATM*-deficient PDAC cells depend on ATR-mediated repair, they exhibit an enhanced sensitivity to ATR inhibitors. In preclinical models, combining ATR inhibition with DNA-damaging drugs dramatically increases tumor cell mortality [233,234]. The loss of *ATM* function in PDAC was related to an elevated radiosensitivity, and platinum-based chemotherapy’s effects are enhanced, allowing tumor cells to accumulate DNA damage more effectively [235,236]. A potential has been observed for Synthetic Lethal Approaches Investigational agents combining PARP inhibitors with ATR inhibitors in *ATM*-deficient settings, because dual inhibition causes synergistic cytotoxicity and interferes with the complementary repair process [237,238].

### 7.4. CHEK1/2

The *CHEK1* gene encodes a serine/threonine kinase that is activated in the *ATR* pathway in response to replication stress and DNA damage. To maintain genome integrity, it is essential for stabilizing replication forks, starting cell cycle arrest at the S/G2 checkpoints, and coordinating DNA repair by acting at replication origins and stalled forks [239,240]. Following DNA DSB, *CHEK2* phosphorylates important substrates, such as p53 and *CD25* phosphatases, after *ATM* activation. In the presence of DNA damage, these events slow down the progression of the cell cycle by reinforcing the G1/S and G2/M checkpoints [241,242]. In DNA damage signaling, *CHEK1* and *CHEK2* are key effectors. *CHEK1* primarily controls replication stress and S/G2 regulation, while *CHEK2* initiates the DNA damage response after DSB. Together, they enforce high-fidelity DNA repair and stop mutation accumulation [243].

In a study performed on 298 PDAC patients tested by multigene panels, *CHEK2* pathogenic variants were observed in approximately 4.1% of patients; however, no *CHEK1* germline variants were detected [216]. In a larger Mayo Clinic study, *CHEK2* variations accounted for approximately 11% of non-*BRCA/ATM* pathogenic germline mutations, with *CHEK1* absent, suggesting its rarity in PDAC germline predisposition [217]. Moreover, in several systematic reviews, there are no documented germline *CHEK1* mutations in PDAC, showing that it is extremely uncommon as a heritable susceptibility gene in this type of tumor [216,227].

The ATR-CHEK1 checkpoint pathway is essential for buffering the high replication stress brought on by oncogenic *KRAS* mutations in PDAC, which enables tumor cells to survive despite severe DNA damage and replication fork instability [244,245]. Carriers of germline DDR gene mutations, including *CHEK2*, did not show a significantly different OS from non-carriers in a population-based analysis. This implies that other factors might have a greater predictive influence than *CHEK2* [246]. No significant survival associations have been found, since germline *CHEK1/2* mutations in PDAC are not common [227,247].

The use of DNA-damaging chemotherapy in DDR-mutated genes is supported by the improved outcomes observed with FOLFIRINOX/platinum-containing regimens for tumors with DDR gene mutations, which include *CHEK2* [247]. Since *CHEK2*-mutated PDAC only has anecdotal evidence of PARP sensitivity, PARP use in *CHEK2* carriers should be individualized. Inhibiting *CHEK1*, or the upstream *ATR-CHEK1* axis, can be synthetically lethal under high replication stress conditions, especially when combined with *EKR* pathway inhibition, as preclinical *KRAS*-mutant PDAC models show [244,248,249].

### 7.5. Summary

In addition to the summary in Table 2, we wanted to incorporate two recently published studies.

A study performed by Kryklyva et al. examined germline pathogenic variants (gPVs) in 473 PDAC patients who also had at least one extra-pancreatic malignancy based on a comprehensive national pathology registry. In 16% of patients, pathogenic variants were found, particularly in *ATM* (22 cases), *BRCA2* (10 cases), and *CHEK2* (10 cases). It has also been observed that the frequency of gPVs was especially high in PDAC patients with concurrent ovarian cancer (40%), melanoma (28%), and gastric cancer (22%). These results support focused germline testing efforts by underscoring the fact that a person’s broader cancer history significantly increases the likelihood of revealing inherited cancer risk variants [250].

The NORPACT-2 study, conducted on a population-based Norwegian cohort, evaluated molecular testing practices among patients with borderline resectable or locally advanced pancreatic cancer (LAPC). Even though worldwide standards indicate molecular profiling for *KRAS* status and/or MSI, just 16% of the 188 consecutive patients treated between 2018 and 2020 underwent this procedure. One patient with MSI-high status among those examined received immunotherapy, underwent surgical resection, and showed a complete pathological response. Testing was more common among patients receiving FOLFIRINOX or managed at a certain hospital trust. This study makes a compelling case for the routine use of genetic testing to find actionable indicators and possibly enhance surgical and therapeutic outcomes, while also highlighting an underuse of molecular diagnostics in potentially resectable PDAC [251].

## 8. Conclusions

Approximately 5–10% of PCs are WT KRAS. In this group, other oncogenic driving mechanisms are more common, including rare, but potentially druggable, gene fusions: ALK, ROS1, NTRK, RET, FGFR, and NRG1. Given the limited PDAC-specific mechanistic evidence, further studies in pancreatic models are needed to validate the oncogenic role of these fusions and provide stronger mechanistic validation. Besides ERBB2 amplification, other genetic amplifications, such as FGF3, FGFR3, NTRK, and MET, await further in-depth study. There is also a group of gene amplifications amenable to known targeted therapies occurring in 10% of WT PC [104]. Genetic mutations should also be considered more broadly, particularly for GNAS and chromatin-remodeling genomic alterations.

Unlike KRAS-mutated PDAC, where the treatment possibilities remain limited, the diversity of WT KRAS tumors offers new opportunities for precision oncology. Molecularly tailored agents have the potential to increase survival outcomes and treatment effectiveness. Emerging concepts such as BRAF/MEK inhibition, bispecific HER2/HER3 antibodies for *NRG1*-driven tumors, or integrating ATR inhibitors in patients with ATM alterations are key aspects for future studies.

Germline mutations also play a significant role in the WT KRAS PDAC subgroup, particularly in the DDR genes. They are predictive of sensitivity to platinum-based chemotherapy and PARP inhibitors, and their diagnosis may have familial consequences. Combining PARP inhibitors with other drugs in PC, particularly for WT KRAS tumors, is a complex but potentially promising area of research. A careful consideration of potential benefits, risks, and ongoing clinical trials is crucial.

In conclusion, there is a high need for comprehensive genome profiling to detect rare aberrations that may be an option for precision oncology approaches in patients with WT KRAS PDAC. Unlike KRAS-mutated PDAC, where targeted treatment options are limited, in WT KRAS, the detection of fusions, amplifications, or mutations can directly pave the way for effective treatment with molecularly targeted inhibitors. Much of the current evidence supporting precision oncology in this setting is derived from case reports and clinical trials on a small patient group; therefore, enrolling patients in clinical trials is critical to generate robust data and enable more definitive treatment recommendations. Nevertheless, considering that these genetic modifications manifest with a prevalence of about 1–3%, performing extensive phase 3 clinical trials is practically impossible. Each documented response adds crucial information about the clinical utility of the drugs and brings us closer to achieving appropriately targeted therapy. Establishing international prospective registries including a database of treatment responses would enable a genuine evaluation of the efficacy of a certain therapy.

### Limitations

First, this review is limited by the scarcity of high-level clinical evidence, the low incidence of molecular alterations, and the heterogeneity of available studies. The conclusions rely on small cohorts, retrospective analyses, and published case reports. Further prospective, multicenter efforts are needed to validate the clinical relevance of these findings. Second, the literature was selected by the authors, which may introduce selection bias despite efforts to include the most significant and current evidence.

## Figures and Tables

**Figure 1 cancers-17-03769-f001:**
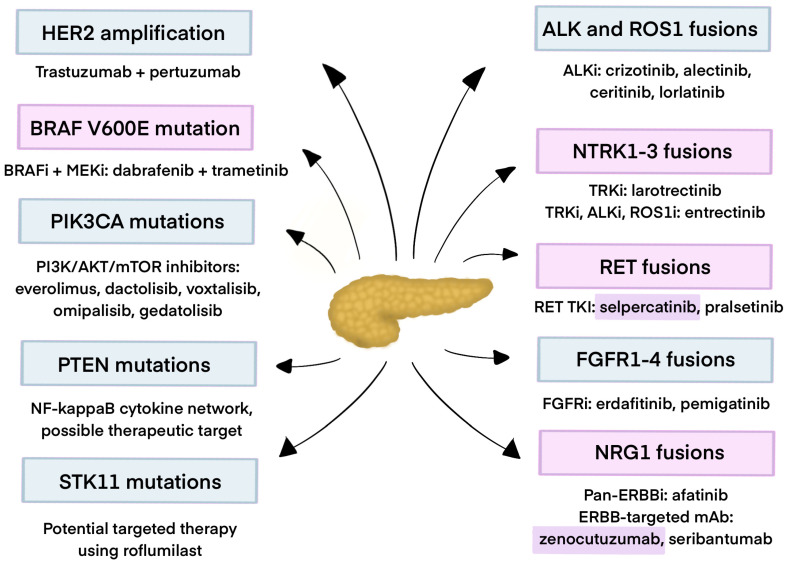
Targetable gene fusions, amplifications, and mutations in KRAS wild-type pancreatic ductal adenocarcinoma. Approved therapies for treating specific alterations are highlighted in purple. Abbreviations: ALK—anaplastic lymphoma kinase; ROS1—c-ros oncogene 1; NTRK—neurotrophic tyrosine receptor kinase; FGFR—Fibroblast Growth Factor Receptor; NRG1—Neuregulin 1; HER2—human epidermal growth factor receptor 2; PI3K—Phosphoinositide 3-Kinase; AKT—Protein Kinase B; mTOR—Mechanistic Target of Rapamycin; RET—Rearranged During Transfection; ERBB—Erythroblastic Oncogene B; TRK—tropomycin receptor kinase; STK11—serine/threonine kinase 11; PTEN—phosphatase and tensin homolog; BRAF—v-Raf Murine Sarcoma Viral Oncogene Homolog B1; MEK—Mitogen-Activated Protein Kinase Kinase.

**Figure 2 cancers-17-03769-f002:**
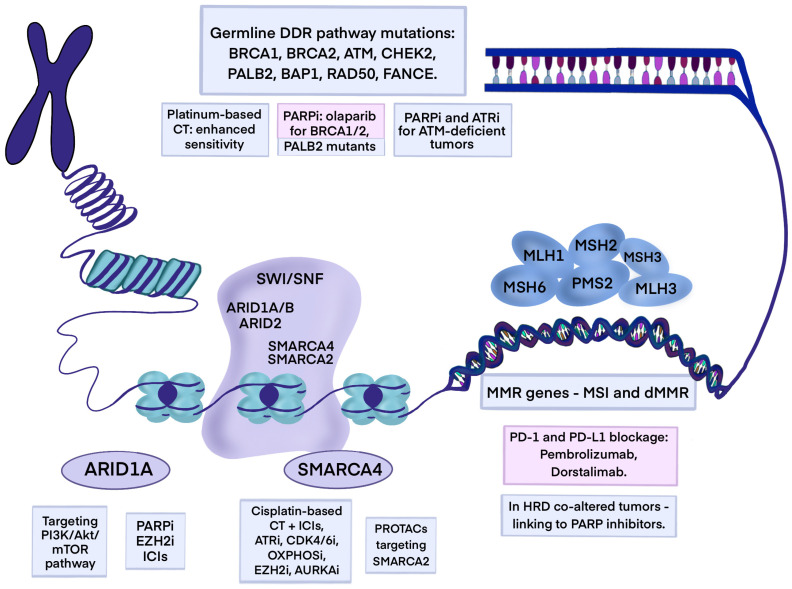
Therapeutic targets for chromatin remodeling genomic alterations, mismatch repair deficiency, microsatellite instability, and germline DNA damage repair pathway mutations in WT-KRAS PDAC. The figure summarizes three molecular vulnerability categories in KRAS–WT PDAC: chromatin remodeling alterations with two therapeutic targets: ARID1A and SMARCA4; mismatch repair deficiency phenotype prone to effective PD-1 blockade; and germline DDR-mutated tumors exhibiting synthetic lethality with PARP inhibition and sensitivity to platinum-based chemotherapy or ATR inhibition. Abbreviations: dMMR—mismatch repair deficient; MSI—microsatellite instability; DDR—DNA damage repair; PARP—poly ADP ribose polymerase; ATR—ataxia telangiectasia and Rad3-related; SWI/SNF—switch/sucrose non-fermentable complex; EZH2—enhancer of zeste homolog 2; ICI—immune checkpoint inhibitor; PI3K—Phosphoinositide 3-Kinase; AKT—Protein Kinase B; mTOR—Mechanistic Target of Rapamycin; CT—chemotherapy; SMARCA4—SWI/SNF-Related, Matrix-Associated, Actin-Dependent Regulator of Chromatin, Subfamily A, Member 4; ARID1A/ARID1B—AT-Rich Interaction Domain 1A/1B; PROTAC—Proteolysis-Targeting Chimera; CDK4/6—cyclin-dependent kinase 4/6; AURKA—Aurora Kinase A; HRD—homologous recombination deficiency; PD-1 and PD-L1—programmed death-1 and -ligand 1; BRCA1/2—Breast Cancer Type 1/2 Susceptibility Protein; ATM—ataxia telangiectasia mutated; BAP1—BRCA1-Associated Protein 1; RAD50—RAD50 Double-Strand Break Repair Protein; FANCE—Fanconi Anemia Complementation Group E Protein; PALB2—partner and localizer of BRCA2; CHEK2—checkpoint kinase 2.

**Table 1 cancers-17-03769-t001:** Summary of fusion genes in KRAS-WT PDAC with estimated prevalence, targeted therapies, and key clinical trials. Based on [22,25,33,42,45,55,72,88,89,90,91,92,93].

Fusion Gene	Prevalence in PDAC	Common Fusion Partners	Targeted Therapies	Key Clinical Trials (NCT)
ALK	~0.16% overall; ~1.3% in <50y	*EML4*, *STRN*, *KANK4*	Crizotinib, Ceritinib, Alectinib, Lorlatinib	NCT02227940 NCT02568267
ROS1	≤0.3%	*SLC4A4*, *SLC34A2*, *CENPW*	Crizotinib, Entrectinib, Lorlatinib	NCT02568267
NTRK	~0.3%	*TPR*, *EML4*, *KANK1*, *THAP1*, *SEL1L*, *CTRC*, *NOS1AP*, *ERC1*	Larotrectinib, Entrectinib	NCT02122913, NCT02637687, NCT02576431, NCT02568267
RET	~0.6%	*CCDC6*, *TRIM33*, *TRIM24*, *ERC1*	Selpercatinib, Pralsetinib, BOS172738	NCT03157128, NCT03037385, NCT03780517
FGFR	1–1.5%	There are at least 114 unique *FGFR* fusion partner genes: FGFR2-BICC1, FGFR2-KIAA1217, FGFR2-SORBS1, FGFR2-AHCYL1, FGFR1-TACC1, FGFR3-TACC3	Erdafitinib, Pemigatinib	NCT04083976, NCT03822117
NRG1	<1%	*ATP1B1*, *APP*, *CD74*, *CDH6*, *SARAF*, *ROCK1*	Afatinib, Zenocutuzumab, Seribantumab	NCT02912949, NCT04383210

Abbreviations: ALK—anaplastic lymphoma kinase; ROS1—c-ros oncogene 1; NTRK—neurotrophic tyrosine receptor kinase; FGFR—Fibroblast Growth Factor Receptor; NRG1—Neuregulin 1; RET—Rearranged During Transfection; EML4—Echinoderm Microtubule-Associated Protein-Like 4; STRN—Striatin; KANK1—KN Motif and Ankyrin Repeat Domains 1; KANK4—KN Motif and Ankyrin Repeat Domains 4; SLC4A4—Solute Carrier Family 4 Member 4; SLC34A2—Solute Carrier Family 34 Member 2; CENPW—Centromere Protein W; TPR—translocated promoter region protein; THAP1—THAP Domain-Containing, Apoptosis-Associated Protein 1; SEL1L—SEL1L Adaptor Subunit of ERAD E3 Ubiquitin Ligase; NOS1AP—Nitric Oxide Synthase 1 Adaptor Protein; ERC1—ELKS/Rab6-Interacting/CAST Family Member 1; CCDC6—Coiled-Coil Domain-Containing 6; TRIM33—Tripartite Motif-Containing 33; TRIM24—Tripartite Motif-Containing 24; FGFR2-BICC1—Fibroblast Growth Factor Receptor 2–BicC Family RNA Binding Protein 1 Fusion; FGFR2-KIAA1217—Fibroblast Growth Factor Receptor 2–KIAA1217 Fusion; FGFR2-SORBS1—Fibroblast Growth Factor Receptor 2–Sorbin and SH3 Domain-Containing 1 Fusion; FGFR2-AHCYL1—Fibroblast Growth Factor Receptor 2–Adenosylhomocysteinase-Like 1 Fusion; FGFR1-TACC1—Fibroblast Growth Factor Receptor 1–Transforming Acidic Coiled-Coil-Containing Protein 1 Fusion; FGFR3-TACC3—Fibroblast Growth Factor Receptor 3–Transforming Acidic Coiled-Coil-Containing Protein 3 Fusion; ATP1B1—ATPase Na^+^/K^+^-transporting subunit beta 1; APP—Amyloid beta precursor protein; CD74 - Cluster of Differentiation 74; CDH6—Cadherin 6; SARAF—Store-operated calcium entry-associated regulatory factor; ROCK1—Rho-associated coiled-coil-containing protein kinase 1.

**Table 2 cancers-17-03769-t002:** Summary of germline DNA damage repair pathway mutations in PDAC.

Category	*BRCA1*	*BRCA2*	*PALB2*	*ATM*	*CHEK2*	*CHEK1*
Primary DDR role	Early HRR: damage sensing, end-resection with MRN/CtIP; recruits PALB2, supports RAD51 loading	Late HRR: directly loads RAD52 via BRC repeats to ssDNA	HRR scaffold linings BRCA1 to BRCA1 and RAD51; stabilizes RAD52 filaments; DNA binding	DSB-response kinase; phosphorylates p53, BRCA2, H2AX	Checkpoint kinase downstream of ATM; enforces G1/s and G2/M checkpoints	Checkpoint kinase downstream of ATR; buffers replication stress; S/G2 checkpoints
Germline prevalence in PDAC	Part of combined *BRCA1/2* ~3–7%; population background for *BRCA1* ~0.2%	More frequent than *BRCA1*; ~3–7% combined *BRCA1/2*; ~2/3 of *BRCA*-related PDAC	~0.5–1.5%	~2–3% unselected PDAC, ~2.38% in genome-first population analysis	~4.1% in a 298-patient series; ~11% of non-*BRCA/ATM* gPVs in a familial registry	Extremely rare in PDAC germline cohorts reported
Enrichment in high-risk groups	Enriched in FPC; HRR genes up to 20% in FPC cohorts	Enriched in FPC; *BRCA2* dominant within *BRCA* cases	Present in high-risk settings (as the HRR gene)	Present in high-risk settings (as the HR gene)	Reported on multigene panels	No enrichment signal reported
Testing/diagnosis	Universal germline testing: RAD52 foci and HRD assays may complement selection	Universal germline testing: RAD52 foci and HRD assays may complement selection	Included on multigene panels; autosomal-dominant; cascade testing for relatives	Included in multigene panels; paired tumor-normal often used to confirm biallelic inactivation; IHC ATM loss can flag pathway disruption; CAPS-based MRI/EUS surveillance for carriers with family history; cascade testing	Detected as part of broad DDR panels; ACMG/AMP classification used	Included on panels, but detection is typically incidental due to rarity
Therapy sensitivity/strategies	Platinum sensitivity; PARP maintenance (olaparib)	Platinum sensitivity; PARP maintenance (olaparib); often a stronger signal for platinum response than *BRCA1*	Platinum and PARP sensitivity; reports of rucaparib maintenance	Sensitivity to ATR inhibition, platinum, and radiation	Pan-DDR rationale for platinum; PARP use described anecdotally	Preclinical rationale: ATR-CHEK1 axis targeting under replication stress
Tumor behavior/survival	HRD with increased platinum responsiveness; different survival reported between *BRCA2* and *BRCA1* in one cohort	*BRCA2* carriers showed better OS than *BRCA1*	May mirror *BRCA*-type HRD responsiveness	Can be aggressive with tumoral ATM loss	Population analysis showed no significant mortality difference vs. non-carriers	No PDAC germline survival data reported
References	[182,183,184,185,186,188,189,190,196,198,199,207,208,209,210,211,212,213,214,215,216,217,222,223,224,225,226,227,228,229,230,231,232,239,240,241,242,243,244,245,246,247,248,249]					

Abbreviations: PDAC—pancreatic ductal adenocarcinoma; DDR—DNA damage repair; HRR—homologous recombination repair; MRN—MRE11-RAD50-NBS1 complex; CtIP—C-terminal-Binding Protein-Interacting Protein; RAD51—RAD51 Recombinase; RAD52—RAD52 Homolog; DSB—double-strand break; p53—tumor protein p53; H2AX—H2A Histone Family Member X; FPC—familial pancreatic cancer; HRD—homologous recombination deficiency; IHC—immunohistochemistry; CAPS—Cancer of the Pancreas Screening Consortium; MRI—Magnetic Resonance Imaging; EUS—endoscopic ultrasound; ACMG/AMP—American College of Medical Genetics and Genomics/Association for Molecular Pathology; PARP—poly ADP ribose polymerase; ATR—ataxia telangiectasia and Rad3-related protein; OS—overall survival; gPVs - germline Pathogenic Variants.

## Data Availability

No new data were created or analyzed in this study. Data sharing is not applicable to this article.

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
