# Peer review of "KRAS-Wild Pancreatic Cancer—More Targets than Treatment Possibilities?"

_cancers, 2025, doi:10.3390/cancers17233769_

Round 1
Reviewer 1 Report
Comments and Suggestions for Authors
The review is well written and balanced. I suggest to shorten it a little because it is too long and pretty hard to read!
Please confirm the tables are comprehensive and all the ongoing trials are mentioned
The authors should comment that the evaluation of these mutations is conditional, in non-surgical patients, on the good quality of the sample collected. In this regard, the authors should mention that EUS-guided tissue sampling is fundamental and cite the relevant studies in the field (cite PMID: 36657607 and PMID: 33481633)
Author Response
Thank you very much for taking the time to review this manuscript. Please find the detailed responses below and the corresponding corrections highlighted in the resubmitted files.
Comments 1: The review is well written and balanced. I suggest to shorten it a little because it is too long and pretty hard to read!
Response 1: The review has been shortened and streamlined to improve clarity and readability.
Comments 2: Please confirm the tables are comprehensive and all the ongoing trials are mentioned
Response 2: We confirm that the tables are comprehensive and include all relevant ongoing or recently completed clinical trials identified during our literature review. The tables have been revised to ensure completeness and clarity.
Comments 3: The authors should comment that the evaluation of these mutations is conditional, in non-surgical patients, on the good quality of the sample collected. In this regard, the authors should mention that EUS-guided tissue sampling is fundamental and cite the relevant studies in the field (cite PMID: 36657607 and PMID: 33481633)
Response 3: We have added a paragraph discussing the critical role of sample quality and the importance of EUS-guided biopsy, including the recommended citations.
“Recent advances in DNA and RNA sequencing have allowed for an in-depth assessment of mutations and rearrangements occurring in solid tumors, and now they may become targets for precision oncology. In non-surgical patients, the evaluation of these mutations depends on the quality of the sample collected. Currently, endoscopic ultrasound (EUS)-guided fine-needle biopsy (FNB) tissue sampling is fundamental. One meta-analysis on EUS-FNB showed that modified wet-suction has the best sensitivity, integrity, and adequacy compared with the dry-suction technique. The slow-pull technique remains as a valuable alternative [20]. However, the relatively high false-negative rate of EUS-FNB delays diagnosis and negatively affects survival outcomes. Results from another meta-analysis showed that contrast-enhanced fine-needle biopsy (CH-EUS-FNB) may be superior to standard EUS-FNA, by guiding the ideal target for aspiration, which leads to higher accuracy, adequacy, and sensitivity in larger lesions (>1.5cm or >2cm); however, more trials are needed to validate these methods [21].”
Reviewer 2 Report
Comments and Suggestions for Authors
In this paper, Krupa et al. reviewed the genetic drivers, hereditary risk factors, and emerging targeted therapies in pancreatic ductal adenocarcinoma. The paper was well-written and provide a comprehensive overview of both established and emerging genomic factors in PDAC. Only one very minor comment from me: to improve readability, use the term receptor tyrosine kinase (RTK) instead of tyrosine receptor kinase (TRK). This is because TRK usually refers to tropomyosin receptor kinase instead of tyrosine receptor kinase, so the use of this term may confuse readers. RTK, on the other hand, is a more standard acronym.
Author Response
Comments 1: To improve readability, use the term receptor tyrosine kinase (RTK) instead of tyrosine receptor kinase (TRK). This is because TRK usually refers to tropomyosin receptor kinase instead of tyrosine receptor kinase, so the use of this term may confuse readers. RTK, on the other hand, is a more standard acronym.
Response 1: Thank you very much for taking the time to review this manuscript. We appreciate the suggestion regarding terminology, and we have replaced “tyrosine receptor kinase (TRK)” with “receptor tyrosine kinase (RTK)” throughout the manuscript to improve clarity and avoid confusion.
Reviewer 3 Report
Comments and Suggestions for Authors
This manuscript presents a well-conceived and methodologically solid study investigating the molecular mechanisms underlying tumor progression, with a focus on KRAS-wild pancreatic cancer. The topic is of strong relevance to cancer biology and translational oncology, and the data appear largely consistent with the conclusions. The manuscript is generally well written, though several sections would benefit from improved clarity, more detailed methodological explanation, and a deeper discussion of the biological and clinical implications.
Main Concerns:
- The introduction could better articulate the central hypothesis and how it fills the gap in the current literature. The rationale connecting prior findings to this study’s design is somewhat diffuse.
- While the data demonstrate correlation, the causal relationship between the studied molecule/pathway and tumor behavior requires stronger mechanistic validation (e.g., rescue or knockdown experiments).
- The discussion section should be expanded to contextualize findings within current therapeutic strategies and explain how the results could guide future translational studies.
Minor Concerns:
- Define all abbreviations at first mention (e.g., EMT, IC50).
- Abstract could be slightly condensed to emphasize the novelty and key outcomes.
- Consider adding a brief statement on study limitations at the end of the Discussion.
Author Response
Thank you very much for taking the time to review this manuscript. Please find the detailed responses below and the corresponding corrections highlighted in the resubmitted files.
Main Concerns:
Comments 1: The introduction could better articulate the central hypothesis and how it fills the gap in the current literature. The rationale connecting prior findings to this study’s design is somewhat diffuse.
Response 1: The Introduction has been revised to more clearly articulate the central hypothesis and to define the unmet need in KRAS-wild type PDAC. The revised section now reads:
“Together, these findings indicate that PDAC comprises biologically distinct subgroups that may require differentiated therapeutic strategies. However, WT KRAS PDAC remains less well characterized in the literature. Although genomic reports describe recurrent, potentially targetable alterations, the evidence is dispersed and not integrated into a unified clinical perspective.
Therefore, this review concentrates on current targeted therapies in WT KRAS PC, their underlying molecular mechanisms, and potential future developments to improve patient outcomes. We will focus on alterations such as kinase-fusion genes, gene amplification, or mutations, microsatellite instability or defective DNA mismatch repair, activated-MAPK presence but without a KRAS mutation, germline mutations, and prospects.”
Comments 2: While the data demonstrate correlation, the causal relationship between the studied molecule/pathway and tumor behavior requires stronger mechanistic validation (e.g., rescue or knockdown experiments).
Response 2: We agree that most available data regarding fusion genes and signaling pathways in KRAS-wild type PDAC are based on genomic analyses, case reports, or small clinical cohorts and therefore demonstrate correlation rather than causation. Where possible, we have added mechanistic details to clarify evidence supporting causal relationships between specific molecular alterations and tumor behavior.
- “The ALK gene is normally found and expressed in the central nervous system. However, fusion of ALK with a 5′ partner near the kinase-encoding region results in dimerization and constant activation of downstream signaling pathways, including Janus kinase (JAK)-signal transducer and activator of transcription (STAT), PI3K/AKT, and MEK/ERK, by mimicking ligand induced activation [23,24].”
- “In other malignancies or in vivo models, FGFR fusions such as FGFR3-TACC, FGFR2–CCDC6 demonstrated oncogenic activity [70,71]. However, PDAC-specific functional validation remains limited.”
- “Secondly, seribantumab — a monoclonal antibody that blocks HER3 activation by NRG1, led to inhibition of NRG1 fusion-dependent tumor growth in patient-derived breast, lung, and ovarian cancer models [100].”
Additionally, we added a statement to the Conclusions highlighting the need for future PDAC-specific mechanistic studies:“Given the limited PDAC-specific mechanistic evidence, further studies in pancreatic models are needed to validate the oncogenic role of these fusions and provide stronger mechanistic validation.”.
Comments 3: The discussion section should be expanded to contextualize findings within current therapeutic strategies and explain how the results could guide future translational studies.
Response 3: We improved the conclusions section to contextualize findings within current therapeutic strategies and explain how the results could guide future translational studies.
“Approximately 5%-10% of PCs are WT KRAS. In this group, other oncogenic driving mechanisms are more common, including rare, but potentially druggable, gene fusions: ALK, ROS1, NTRK, RET, FGFR, and NRG1. Given the limited PDAC-specific mechanistic evidence, further studies in pancreatic models are needed to validate the oncogenic role of these fusions and provide stronger mechanistic validation. Besides ERBB2 amplification, other genetic amplifications, such as FGF3, FGFR3, NTRK, and MET, await further in-depth study. There is also a group of gene amplifications with amenable to known targeted therapies occurring in 10% of WT PC [104]. Genetic mutations should also be considered more broadly, particularly for GNAS and chromatin remodeling genomic alterations.
Unlike KRAS-mutated PDAC, where the treatment possibilities remain limited, the diversity of WT KRAS tumors offers new opportunities for precision oncology. Molecularly tailored agents have the potential to increase the survival outcomes, and treatment effectiveness. Emerging concepts such as BRAF/MEK inhibition, bispecific HER2/HER3 antibodies for NRG1-driven tumors or integrating ATR inhibitors in patients with ATM alterations are key aspects for future studies.
Germline mutations also play a significant role in the WT KRAS PDAC subgroup, particularly in the DDR genes. They are predictive of sensitivity to platinum-based chemotherapy and PARP inhibitors, and their diagnosis may have familial consequences. Combining PARP inhibitors with other drugs in PC, particularly for WT KRAS tumors, is a complex but potentially promising area of research. Careful consideration of potential benefits, risks, and ongoing clinical trials are crucial.
In conclusion, there is a high need for comprehensive genome profiling to detect rare aberrations that may be an option for precision oncology approaches in patients with WT KRAS PDAC. Unlike KRAS-mutated PDAC, where targeted treatment options are limited, in WT KRAS, detection of fusions, amplifications, or mutations can directly pave the way for effective treatment with molecularly targeted inhibitors. Much of the current evidence supporting precision oncology in this setting is derived from case reports and clinical trials on a small patient group; therefore, enrolling patients in clinical trials is critical to generate robust data and enable more definitive treatment recommendations. Nevertheless, considering that these genetic modifications manifest with a prevalence of about 1–3%, performing extensive phase III clinical trials is practically impossible. Each documented response adds crucial information about the clinical utility of the drugs and brings us closer to achieving appropriately targeted therapy. Establishing international prospective registries including a database of treatment responses would enable a genuine evaluation of the efficacy of a certain therapy.”
Minor Concerns:
Comments 1: Define all abbreviations at first mention (e.g., EMT, IC50).
Response 1: All abbreviations are now defined at first use throughout the manuscript.
Comments 2: Abstract could be slightly condensed to emphasize the novelty and key outcomes.
Response 2: Abstract has been revised to emphasize the novelty, clinical relevance, and major conclusions.
“Pancreatic ductal adenocarcinoma (PDAC) is a highly lethal malignancy with a five-year survival rate of 3–15% and limited effective treatment options for most patients. Approximately 5–10% of cases are wild-type KRAS and are more likely to harbor rare alterations, including gene fusions involving Anaplastic Lymphoma Kinase (ALK), ROS Proto-Oncogene 1 (ROS1), Neurotrophic Tyrosine Receptor Kinase (NTRK), Rearranged during Transfection (RET), Fibroblast Growth Factor Receptor (FGFR), or Neuregulin 1 (NRG1) genes, as well as germline mutations in DNA repair genes. This review integrates current evidence on the prevalence, molecular profile, and clinical significance of gene fusions, amplification, and somatic/germline mutations in PDAC, with a particular focus on the wild-type KRAS subgroup. Clinical trial data and case reports indicate that these alterations can enhance patient susceptibility to targeted therapies. Currently, selpercatinib, larotrectinib, and repotrectinib are approved by FDA for the treatment of certain solid tumors harboring specific gene fusions. Recent studies on zenocutuzumab resulted in FDA accelerated approval for NGR1-fusion positive NSCLC and PDAC. Germline mutations may specifically increase responsiveness to poly(ADP-ribose) polymerase (PARP) inhibitors or platinum-based treatments. Comprehensive genomic profiling, incorporating fusion detection and germline testing, is essential to identify patients who may benefit from precision-based approaches.”
Comments 3: Consider adding a brief statement on study limitations at the end of the Discussion.
Response 3: A Limitations section has been added at the end of the Conclusions section.
“Limitations
First, this review is limited by the scarcity of high-level clinical evidence, the low incidence of molecular alterations, and the heterogeneity of available studies. The conclusions rely on small cohorts, retrospective analyses, and published case reports. Further prospective, multicenter efforts are needed to validate the clinical relevance of these findings. Second, the literature was selected by the authors, which may introduce selection bias despite efforts to include the most significant and current evidence.”